# Regeneration of Two-Walled Infrabony Periodontal Defects in Swine After Buccal Fat Pad-Derived Dedifferentiated Fat Cell Autologous Transplantation

**DOI:** 10.3390/biom15040604

**Published:** 2025-04-20

**Authors:** Daisuke Akita, Naoki Tsukimura, Tomohiko Kazama, Rie Takahashi, Yoshiki Taniguchi, Jin Inoue, Ayana Suzuki, Nodoka Tanabe, Keisuke Seki, Yoshinori Arai, Masatake Asano, Shuichi Sato, Yoshiyuki Hagiwara, Koichiro Kano, Masaki Honda, Taro Matsumoto

**Affiliations:** 1Department of Partial Denture Prosthodontics, Nihon University School of Dentistry, Tokyo 101-8310, Japan; hagiwara.yoshiyuki@nihon-u.ac.jp; 2Division of General Dentistry, The Nippon Dental University School of Life Dentistry, Tokyo 102-8158, Japan; tsukimura@tky.ndu.ac.jp; 3Division of Cell Regeneration and Transplantation, Department of Functional Morphology, Nihon University School of Medicine, Tokyo 173-8610, Japan; kazama.tomohiko@nihon-u.ac.jp (T.K.); matsumoto.taro@nihon-u.ac.jp (T.M.); 4Section of Laboratory Animals, Nihon University School of Medicine, Tokyo 173-8601, Japan; takahashi.rie@nihon-u.ac.jp (R.T.); taniguchi.yoshiki@nihon-u.ac.jp (Y.T.); 5Division of Applied Oral Sciences, Nihon University Graduate School of Dentistry, Tokyo 101-8310, Japan; deji22005@g.nihon-u.ac.jp (J.I.); deay23016@g.nihon-u.ac.jp (A.S.); deno24016@g.nihon-u.ac.jp (N.T.); 6Department of Comprehensive Dentistry and Clinical Education, Nihon University School of Dentistry, Tokyo 101-8310, Japan; seki.keisuke@nihon-u.ac.jp; 7Department of Oral and Maxillofacial Radiology, Nihon University School of Dentistry, Tokyo 101-8310, Japan; arai.yoshinori@nihon-u.ac.jp; 8Department of Pathology, Nihon University School of Dentistry, Tokyo 101-8310, Japan; asano.masatake@nihon-u.ac.jp; 9Department of Periodontology, Nihon University School of Dentistry, Tokyo 101-8310, Japan; satou.shuuichi@nihon-u.ac.jp; 10Laboratory of Cell and Tissue Biology, College of Bioresource Science, Nihon University, Fujisawa 252-0880, Japan; kano.kouichirou@nihon-u.ac.jp; 11Department of Oral Anatomy, Aichi Gakuin University School of Dentistry, Nagoya 464-8650, Japan; honda-m@dpc.agu.ac.jp

**Keywords:** periodontal tissue regeneration, dedifferentiated fat cells (DFAT cells), transplantation, two-walled infrabony periodontal defect

## Abstract

Mature adipocyte-derived dedifferentiated fat (DFAT) cells show proliferative capabilities and multipotency. Given that the buccal fat pad (BFP) serves as a readily available resource for DFAT cell isolation, BFP-derived DFAT (BFP-DFAT) cells are a promising candidate in orofacial tissue engineering. In this research, we assessed the regenerative capacity of the periodontium through autologous BFP-DFAT cell transplantation in adult swine (micro-minipigs; MMPs). The BFP-DFAT cells were transplanted into inflammation-inducing two-walled infrabony periodontal defects located on the mesial of the second mandibular premolar (*n* = 6). Twelve weeks post-transplantation, a remarkable attachment gain was noted in the DFAT group, based on probing depths and clinical attachment levels. Histological and immunohistochemical analyses indicated new continuous cellular cementum and alveolar bone formation within the created infrabony defect. Well-organized periodontal ligament-like fibers were embedded between newly formed cementum and the alveolar bone. Histometric analysis demonstrated that the DFAT group had a 2.2-fold increase in new alveolar bone length and a 2.2-fold enhancement in vascularization than those in the control group. Except for minor inflammation in the lungs, no teratomas were detected in the recipient MMPs. BFP-DFAT cells significantly enhanced periodontal tissue regeneration, thus representing an optimal source for tissue engineering applications in dentistry.

## 1. Introduction

Periodontal disease, the most prevalent form of inflammation that compromises the supporting structures of the teeth, leads to heightened tooth mobility and eventual tooth loss [1]. Odontotherapy addresses local inflammation and plaques through scaling and root planing, preventing further aggravation and subsequent functional disorders such as mastication, pronunciation, esthetic appreciation, and sensation [2,3]. Periodontal disease generally compromises the structural integrity of periodontal tissues, which encompass the alveolar bone, cementum, and periodontal ligament, and reconstructing their normal structure and function is often challenging [4]. The primary objective of periodontal disease management is to restore the strength and function of the periodontal ligament, cementum, and alveolar bone [5]. Currently, a comprehensive periodontal regenerative treatment is unavailable. Therefore, research into periodontal regeneration is underway to develop new therapeutic strategies.

Recent advancements in stem cell research has resulted in the formulation of therapeutic approaches aimed at addressing diseases through the transplantation or administration of cells that have been cultured in vitro [6,7]. Recent developments in regenerative strategies have yielded significant advantages in dentistry, facilitating the restoration of functional tissues through the transplantation of cells capable of differentiating into these particular tissues. Stem cell-based tissue engineering is an innovative strategy for the regeneration of orofacial structures, particularly periodontal tissues. A diverse array of cell types is employed in tissue engineering, and the most commonly utilized cell types for periodontal regeneration purposes are mesenchymal stem cells (MSCs). Numerous research groups have established that MSCs sourced from bone marrow, periodontal ligament, and alveolar periosteum are successful in enhancing periodontal regeneration, especially in medium- to large-sized animal models [8,9,10,11,12,13,14,15,16,17]. However, these procedures are invasive for donors, and only a limited volume can be obtained. An optimal cellular source for regenerative dentistry should exhibit traits such as pluripotency, a high capacity for proliferation, elevated purity, and high convenience of extraction by dental professionals. Numerous research groups have established that adipose-derived stem/stromal cells (ASCs) are pluripotent [18], and they are used in tissue engineering, including periodontal regeneration [19,20,21]. Although ASCs can be obtained through minimally invasive techniques, they represent a diverse population of cells that encompass multiple stages of differentiation. This heterogeneity can result in inconsistencies in both research findings and transplantation results [22].

In our earlier endeavors, we successfully crafted lipid-free fibroblast-like cells, referred to as dedifferentiated fat (DFAT) cells, via the asymmetrical division of mature adipocytes in ceiling culture methods. This approach utilizes the inherent mature adipocytes, thereby obviating the necessity for the introduction of specific factors [23]. These floating cells found in ceiling cultures comprise almost all lipid-filled mature adipocytes, indicating that DFAT cells originate from a highly homogeneous cell population [24]. Instead of losing mature adipocyte-specific markers, DFAT cells have strong proliferative, adipogenic, and osteogenic capabilities; possess minimal donor age requirements; and demonstrate low immunogenicity [24,25,26]. Our earlier investigations have revealed that the osteogenic potential of DFAT cells isolated from the abdominal subcutaneous fat tissue of rats was higher than that of ASCs in vitro [27,28]. Allogeneic transplantation of DFAT cells has demonstrated a superior capacity for promoting periodontal tissue regeneration in vivo compared to ASCs [28]. Our supplementary research indicated that transplanting autologous DFAT cells harvested from the abdominal subcutaneous adipose tissue of adult swine (micro-minipigs; MMPs) resulted in enhanced attachment gain within the regenerated periodontium. This improvement was characterized by the integration of collagen fibers into the newly developed cementum and bone within inflammation-induced furcation defects in adult miniature pigs [29]. These findings suggest that DFAT cell transplantation could represent a viable therapeutic approach for periodontal regeneration. Nonetheless, the procurement of abdominal subcutaneous adipose tissue poses challenges for dental professionals specializing in oral and maxillofacial surgery.

Nestled snugly between the masseter and buccinator muscles, the buccal fat pad (BFP) finds its home alongside the ascending ramus of the mandible and the zygomatic arch [30]. Its anatomical location renders it both convenient and accessible for dental practitioners, who can obtain it through a straightforward surgical procedure performed under local anesthesia [31]. Over the past four decades, the BFP has been utilized as a self-sourced graft for addressing small to medium-sized maxillofacial imperfections associated with congenital oroantral and/or oronasal conditions [32], the repair of congenital cleft palates [33], oral submucous fibrosis [34,35], intraoral malignant defects [36], and buccal mucosa defects [32,37]. Human ASCs isolated from the BFP are used for innovative bone tissue engineering both in vitro [38] and in vivo [39]. Additionally, a recent investigation indicated that the ability of osteoblastic differentiation in human DFAT cells surpasses that in ASCs obtained from the BFP [40].

Therefore, our objective was to explore the regenerative capabilities of BFP-derived DFAT (BFP-DFAT) cells by transplanting them into two-walled infrabony periodontal defects on the mesial root of the second premolar and examine their safety in relation of MMPs.

Next, we set the null hypothesis that there is no difference in the effects of BFP-DFAT cell transplantation. Conversely, we set the alternative hypothesis that there is an effect from BFP-DFAT cell transplantation.

Specifically, in addition to clinical evaluations such as probing depth (PD) and clinical attachment level (CAL), we examined the effectiveness of BFP-DFAT cell autologous transplantation through histological and immunohistochemical analyses and evaluations. The significance level was established at either 1% or 5%. If the test results show a significant difference, the null hypothesis is rejected.

Furthermore, we assessed safety based on the presence or absence of teratoma formation in major organs.

## 2. Materials and Methods

### 2.1. Experimental Animals

All the animal experiments conducted in this research received approval from the Animal Research and Care Committee of Nihon University (AP19MED014-3 and AP24MED020-1). MMPs utilized in the research were procured from Fuji Micra Inc. (Yamanashi, Japan) through Tokyo Laboratory Animals Science Co., Ltd. (Tokyo, Japan). In accordance with prior scholarly investigations, six healthy adult MMPs, averaging 28.8 ± 2.5 months in age and weighing 28.5 ± 1.2 kg, were acquired and assessed for their overall and oral health prior to the commencement of the experiment (Figure 1a) [29].

### 2.2. Preparation of BFP-DFAT Cells

DFAT cells were derived from adipose tissue by utilizing a modified protocol based on previously established methodologies [24]. In summary, approximately 1 g of the BFP was extracted from the buccal region of MMPs (Figure 1b); the extracted tissue was thoroughly rinsed with phosphate-buffered saline (PBS; Wako, Osaka, Japan), finely chopped, and subjected to digestion using a 0.1% (*w*/*v*) collagenase solution (C6885; Sigma-Aldrich, St. Louis, MO, USA) at 37 °C for 0.5 h with gentle agitation throughout the process. Following this process, the mixture was filtered and centrifuged at 700 rpm for a duration of 1 min, allowing for the collection of the buoyant primary mature adipocytes from the upper layer. Subsequently, mature adipocytes were washed three times with PBS, and approximately 5 × 10^4^ cells were then transferred to flasks designated for ceiling culture (iP-TEC^®^; SANPLATEC, Osaka, Japan), which were filled entirely with Dulbecco’s modified Eagle’s medium (DMEM; Gibco, NY, USA) enriched with 20% fetal bovine serum (FBS; Sigma-Aldrich, Lot 14A189) and incubated at 37 °C in 5% CO_2_. Mature adipocytes exhibited buoyancy, resulting in their adherence to the upper inner surface of the flasks (Figure 2a). Throughout the process of ceiling culture, DFAT cells that resemble fibroblasts were derived from the adhered adipocytes, leading to the formation of cellular colonies (Figure 2b). One week later, the flasks were turned upside down, and the culture medium was replaced with DMEM enriched with 20% FBS, ensuring that the cells settled at the base of the flasks. The culture medium was refreshed every four days until the cells achieved subconfluence (Figure 2c). BFP-DFAT cells were obtained at the third culturing passage, and 1 × 10^6^ BFP-DFAT cells were subsequently placed onto a square collagen matrix (Colla Tape; Zimmer Dental Inc., Carlsbad, CA, USA), meticulously trimmed to about 5 × 10 × 0.3 mm^3^, immersed in the culture medium, and prepared for transplantation (Figure 2d).

### 2.3. In Vivo Experimentation

The in vivo experimentation was conducted in accordance with previously established protocols, with minor modifications, such as extracting if the crowns of primary teeth were present or changing the cleaning methods by using sponge brushes and oral care wet wipes (Asahi Group Foods Company, Tokyo, Japan) [29,41,42]. After emplacement, all the MMPs were housed in a research animal facility at the Nihon University School of Medicine (Tokyo, Japan) while adhering to the regulations set by the Animal Research and Care Committee at Nihon University. All in vivo surgical interventions were carried out under general anesthesia, which was achieved through the intravenous administration of isoflurane (Pfizer Inc., NY, USA) and midazolam (Sandoz K.K., Tokyo, Japan) supplemented with a local anesthetic comprising 2% lidocaine hydrochloride and 1:80,000 epinephrine (Nipro Co., Ltd., Osaka, Japan). Nearly all calculi from the premolars located in the mandibular region in all MMPs were removed by utilizing a scaler (Hu-Friedy, Chicago, IL, USA) as the fundamental initial therapy, and inflammation of the periodontium had improved before the surgery (Figure 3a).

About four weeks later, the periosteum enveloping the buccal surface of the lower jaw was excised following incisions made along the upper edge of the premolars on either side. The mandible alveolar bone situated above the mesial root of the second premolar was subsequently resected by utilizing a Zekrya surgical bur (Siya Medical Equipment Co., Ltd., China) under water injection. The two-walled infrabony periodontal defects were approximately 5 mm in height, 4 mm in width, and extended 3 mm in depth into the mesial root in the 2nd premolar (Figure 3b). A silicone rubber impression material, known as EXAFINE (GC Dental Product Co., Ltd., Tokyo, Japan), was applied to the defect to promote persistent inflammation, inhibit natural bone repair, and facilitate biofilm formation. Subsequently, the flaps were carefully adjusted and fastened using artificial dissolvable sutures (VICRYL; Ethicon Inc., NJ, USA), and the resulting defects were subjected to blind randomization via computer algorithms and subsequently allocated to either the control group or the BFP-DFAT treatment sites.

Four weeks post-surgery, clinical evaluations of PD and CAL were conducted on the mesial root of the second premolar by utilizing a periodontal probe (Sun dental Co., Ltd., Osaka, Japan) based on previous study [15,16,29,43,44]. Following this, buccal intrasulcular incisions were executed to reveal the infrabony periodontal defects. After thorough periodontal debridement, the defects associated with the second premolar were meticulously treated with a scaler and subsequently rinsed with saline. The right and left sides of the subjects were randomly allocated to receive either a collagen matrix alone (*n* = 6) or a collagen matrix infused with BFP-DFAT cells (*n* = 6). The matrices, with or without BFP-DFAT cells, were subsequently placed into the defects and were covered with a membrane (BIOMEND; Zimmer Dental Inc.; Figure 3c,d). The surgical flaps were then sutured using antibacterial artificial dissolvable sutures (VICRYL Plus; Ethicon, Inc.).

Twelve weeks post-transplantation, the evaluations of PD and CAL were conducted once more prior to bilateral mandible extraction, as depicted (“Post-transplantation”; Figure 3e,f) based on previous study [15,16,29,43,44]. The mandible, along with other organs such as the heart, kidney, liver, and lung of MMPs, was excised and subsequently fixed in 10% neutral-buffered paraformaldehyde for 24 h to investigate teratoma formation. All six MMPs underwent a uniform clinical protocol and evaluation procedure.

### 2.4. Histological and Immunohistochemical Staining and Analysis

The specimens underwent decalcification in a 10% solution of ethylenediaminetetraacetic acid (EDTA) for 5 weeks. Following this process, the specimens underwent dehydration via a sequential series of ethanol solutions before being embedded in paraffin. Sagittal plane sections, each measuring 5 µm in thickness, were produced using a microtome, and the paraffin-embedded sections of the second premolar were subsequently stained with hematoxylin and eosin (H&E) as well as azan to observe the periodontal defect. Each organ was stained using H&E in a similar manner without decalcification to observe teratoma formation. For the purpose of immunohistochemistry, the paraffin-embedded specimens were subjected to deparaffinization and rehydration through a series of xylene and ethyl alcohol treatments. Following the inhibition of endogenous peroxidase activity, the slides underwent three rinses in PBS, each lasting 5 min. The runx2 (Abcam, Cambridge, MA, USA), osteocalcin (Takarabio, Shiga, Japan), cathepsin K (Abcam), or periostin primary antibody (Abcam) was diluted at 1:200 or 1:100 and incubated with the samples overnight at a temperature of 4 °C. Following this incubation period, the samples were washed and then incubated at room temperature for an hour. Subsequently, the samples underwent three additional rinses in PBS, each lasting 5 min. To enhance signal detection, the samples were treated with a polymer reagent (EnVidion Plus; Dako, Tokyo, Japan) for 30 min at room temperature to facilitate signal detection. Imaging was performed using a fluorescence microscope, namely the BZ-X710 (KEYENCE, Osaka, Japan).

For the purpose of quantitative analysis, three azan-stained sections from each sample were selected. A pathologist utilized a fluorescence microscope (BZ-H3; KEYENCE, Osaka, Japan) to measure the length and width of the newly formed cementum, assess the degree of neovascularization, and count the number of cementoblasts present in the defects.

### 2.5. Statistical Analysis

The data are presented as means and standard deviations (SDs) for each group. Statistical analyses were conducted utilizing the Excel statistical program file software (ystat2008.xls; Igakutosho-shuppan Ltd., Tokyo, Japan). The Mann–Whitney U test, with or without Bonferroni correction, was utilized for comparisons between two groups or across multiple groups. A significance threshold of *p* < 0.01 and *p* < 0.05 was set to ascertain statistical significance.

## 3. Results

### 3.1. Establishment of BFP-Derived DFAT Cells and Clinical Assessments in the Adult MMPs

BFP-DFAT cells were generated through the asymmetrical division of mature adipocytes sourced from the adipose tissue by employing a ceiling culture technique that leverages the floating ability of adipocytes (Figure 2a–c). The third passage of the 1 × 10^6^ BFP-DFAT cells loaded with the collagen matrix was used for transplantation (Figure 2d).

Following the initial surgical procedure, two-walled infrabony periodontal defects that induce inflammation were successfully established bilaterally on the mesial root of second premolars of the mandible (Figure 3). Figure 4 illustrates the clinical parameters associated with each MMP. In the DFAT group, notable decreases in PD were recorded between the transplantation and post-transplantation phases (Figure 4a). Furthermore, a statistically significant difference was identified between the control and DFAT groups after transplantation (*p* = 0.0344). Similarly, significant decreases in CAL were noted between the transplantation and post-transplantation groups (Figure 4b), with a significant difference also noted between the control and DFAT groups post-transplantation (*p* = 0.0463). Figure 4c presents the average values and SDs of PD and CAL at transplantation and post-transplantation.

### 3.2. Transplantation of BFP-Derived DFAT Cells Enhanced Periodontal Regeneration

Figure 5 displays the photomicrographs of azan-stained mesial roots of the second premolar in both the control and BFP-DFAT groups. The presence of artificially induced infrabony periodontal defects, as well as notch-shaped markings resulting from the surgical procedure, was noted in both experimental groups.

In the control group, disorganized epithelial structures were noted above the cavity (Figure 5a), alongside the formation of hard tissue on the exposed dentin surface. Conversely, in the BFP-DFAT group, orderly epithelium was observed above the cavity, accompanied by the observation of continuous hard tissue formation on the exposed dentin surface. Moreover, an enhanced alveolar bone-like structure was observed above the notch-shaped markings (Figure 5b). The magnified views of the dotted frames within the defect region are illustrated in Figure 6. At a higher magnification of the upper region, the control group exhibited only cement-like hard tissue (Figure 6a). In contrast, the BFP-DFAT group displayed numerous collagen bundles situated between the cement-like hard tissue and the newly formed alveolar bone (Figure 6b). At a higher magnification of the central area, vascularization was evident in all the samples (Figure 6c,d). Notably, on the BFP-DFAT side, newly formed collagen bundles were embedded within the newly developed bone-like structures (Figure 6d). At a higher magnification of the bottom region, photomicrographs stained with H&E revealed a hematoxylin-positive cellular hard tissue structure on the exposed dentin in both groups (Figure 6e,f). On the BFP-DFAT side, continuous collagen bundles embedded in newly formed cellular hard tissue structures and vascularization were visualized (Figure 6f).

The findings from the histometric analysis are presented in Figure 7. In the BFP-DFAT group, the newly formed cementum measured an impressive 2.2 times longer than that in the control group (*p* = 0.0068; Figure 7a). Additionally, the width of the newly formed cementum in the BFP-DFAT group was about 1.7 times greater than that in the control group (*p* = 0.0110; Figure 7b). Figure 7c provides a summary of the average values and SDs of the length and width in the control and BFP-DFAT groups. Furthermore, the extent of neovascularization in the BFP-DFAT group was approximately 2.2 times greater than that in the control group (*p* = 0.0068; Figure 7d). Figure 7e presents a summary of the average values and SDs of neovascularization observed in the control and BFP-DFAT groups. The quantity of newly generated cementoblasts in the BFP-DFAT group was found to be approximately 2.4 times greater than that in the control group (*p* = 0.0394; Figure 7f). Additionally, Figure 7g provides a summary of the average values and SDs of cementoblasts in the control and BFP-DFAT groups.

At an increased magnification of the notch region in the BFP-DFAT group, the photomicrographs stained with H&E clearly delineated the cavity in the artificially created defect site (Figure 8a). The presence of well-organized connective tissue, characterized by neovascularization, was evident between the newly formed alveolar bone and cementum. Furthermore, at an increased magnification of the dotted frame in the notch area of the BFP-DFAT group, immunohistochemical analysis revealed osteoblasts that were positive for RUNX2 and osteocalcin lining the newly formed alveolar bone (Figure 8b,c). Multinucleated osteoclasts that were positive for cathepsin K were identified in proximity to the newly formed alveolar bone (Figure 8d). Moreover, periostin-positive ligaments were prominently observed at an increased magnification within the delineated area of the notch area in the BFP-DFAT group (Figure 8e).

To investigate the formation of teratomas, organs were extracted and stained with H&E. Teratomas were absent in the internal organs, specifically the heart, kidneys, liver, and lungs; however, there was a tendency for capillary congestion in the lung interstitium, with mild lymphocyte aggregation observed in the surrounding area (Figure 9a–d).

## 4. Discussion

We previously reported that when DFAT cells derived from subcutaneous adipose tissue were allogeneically transplanted into periodontal fenestration defects in inbred rats, they not only regenerated the periodontal tissue, but the fluorescently labeled DFAT cells were also observed within the regenerated tissue [28]. Furthermore, when DFAT was autotransplanted into the furcation defects of MMPs, enhanced periodontal regeneration was observed [29]. Therefore, it is conceivable that DFAT cells could serve as a valuable cellular source for periodontal tissue regeneration. Given the challenges faced by dentists and oral surgeons in acquiring subcutaneous fat tissue, this study sought to elucidate the effects of autologous transplantation of BFP-DFAT cells of MMPs in vivo. The objective was to further translational research with a focus on potential clinical applications.

In the present investigation, adipose tissue obtained from the BFP of MMPs was isolated following the established protocol (Figure 1). The findings confirmed that the isolated mature adipocytes engage in asymmetric division, resulting in the generation of fibroblasts (Figure 2). The guidelines recommend that the safety and quality of animal-derived materials, including FBS utilized in stem cell research, must be rigorously assessed [45]. In the present cell culture, FBS that had undergone appropriate quality testing and was approved according to standards was used. As a result, no mycoplasma or viral contaminants were detected in the serum used; however, there is a need to establish a safer cell culture process in the future. Then, we used commercially available collagen sheets as scaffold materials from the perspectives of ensuring that there is no cytotoxicity to the transplanted cells, that the cellular functions such as adhesion, proliferation, and differentiation of the transplanted cells are not inhibited, that they can form complexes with a large number of cells, and that they maintain their mechanical properties until they are appropriately absorbed after transplantation. When actually transplanting the BFP-DFAT cells, we folded the collagen sheets as shown on the left side of Figure 2d and transplanted them into the defect site.

The periodontal tissues of MMPs show many similarities to those of humans, and it has been reported that by the age of 16 months, MMPs exhibit periodontal disease characterized by symptoms such as gum swelling, plaque accumulation, tartar formation, and bleeding upon probing, accompanied by the resorption of alveolar bone [43,46]. The MMPs presented exhibited an unfavorable oral condition characterized by significant calculus accumulation, suggesting a potential predisposition to periodontal disease. Consequently, we implemented fundamental initial therapy for the premolars in all the MMPs before surgery to standardize their oral environment (Figure 3). In this investigation, following the application of a collagen sheet, we subsequently applied a membrane to maximize the retention of BFP-DFAT cells at the defect site, thereby promoting tissue regeneration. Nonetheless, the techniques associated with cell transplantation, particularly concerning the physical characteristics of collagen and the functionality of the membrane, continue to present significant challenges. The results of the clinical parameters suggested that BFP-DFAT cell transplantation is clinically effective against periodontal disease (Figure 4). Additionally, the overall histological analysis indicated that the collagen sheet can maintain space as a scaffold in the BFP-DFAT group (Figure 5). High-magnification histological images revealed a significant presence of collagen fibers and capillaries situated between the newly formed bone and cementum (Figure 6). It is plausible that the transplanted BFP-DFAT cells facilitate tissue regeneration through their differentiation into the cellular components of periodontal tissue. However, in recent years, the importance of the nutritional effect of transplanted MSCs secreting cytokine factors, the extracellular matrix, and exosomes to activate tissue stem cells and progenitor cells present around the defect has been reported [47,48]. Therefore, it is conceivable that the BFP-DFAT cells, by utilizing the space-forming ability of the scaffold material to secrete body fluid factors, activated the stem cells around the defect, resulting in the induction of ideal periodontal tissue regeneration (Figure 7). In the context of immunohistochemical analysis, the presence of osteoblasts and osteoclasts surrounding the alveolar bone indicates a gradual progression towards the formation of newly developed alveolar bone (Figure 8). No teratomas were observed in the major organs after cell transplantation, but mild lymphocyte aggregation was noted in the lung interstitium (Figure 9). The causal relationship between these inflammatory findings and cell transplantation is unclear, but the effects of intubation during general anesthesia may be a consideration. Future translational research may require more rigorous safety verification, including the health status and breeding environment of the purchased swine.

In a translational study conducted by Takedachi and others affiliated with Osaka University, commercially available carbonate apatite was selected as the scaffold material for the autologous transplantation of adipose tissue-derived multi-lineage progenitor cells in canine [49]. Additionally, after observing the expression of IGFBP-6, HGF, and VEGF in vitro, the evaluation of periodontal tissue regeneration was performed through the quantification of hard tissue using micro-computed tomography, in addition to histological analysis.

In our limited translational investigation, we demonstrated clinical significance for the two-walled infrabony periodontal defects, despite the fact that MMP was not administered and immunosuppressants and dental cleaning could not be performed post-transplant; however, the selection of scaffold materials that not only facilitate or enhance the proliferation, differentiation, and trophic factor secretion of BFP-DFAT cells but also exhibit operability and spatial formation capabilities presents significant challenges for future research. Additionally, the establishment of a more rigorous cell transplantation protocol is crucial for the progression of this field.

## 5. Conclusions

The objective of this research was to evaluate the periodontal regenerative capacity and safety of autologous BFP-DFAT cell transplantation in inflammation-induced two-walled infrabony periodontal defects in MMPs. The results indicated that the BFP-DFAT group exhibited significant improvements, including attachment gain, the formation of newly developed cellular cementum, well-organized periodontal ligaments with vascularization, and the regeneration of the alveolar bone structure. Notably, the BFP-DFAT group demonstrated a significant increase in both the length and width of the cementum, as well as an enhancement in neovascularization and an elevated number of newly formed cementoblasts, in comparison to the control group. The periodontal tissue that was reactivated in the BFP-DFAT group exhibited a morphology akin to its original configuration. Importantly, the internal organs of the recipient MMPs showed no signs of teratomas. Given that the BFP can be easily harvested and is readily available in adequate quantities through oral surgical procedures, BFP-DFAT cells, characterized by high purity, robust proliferation, and multilineage differentiation potential, present a valuable cellular resource for applications in periodontal regeneration as well as in orofacial tissue engineering.

## Figures and Tables

**Figure 1 biomolecules-15-00604-f001:**
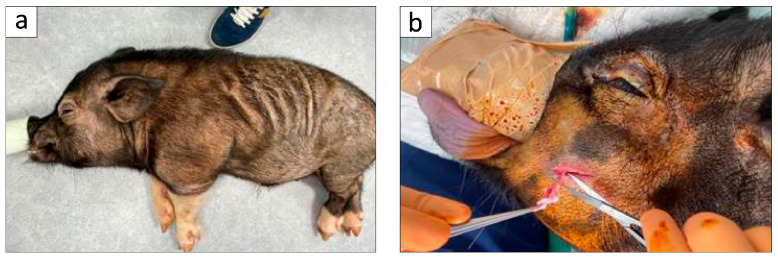
Pre-inclination of adipose tissue harvesting. (**a**) An MMP under inhalational anesthesia at the Nihon University School of Medicine. (**b**) The extraction of approximately 1 g of adipose tissue from the buccal fat pad for the purpose of isolation of mature adipocytes.

**Figure 2 biomolecules-15-00604-f002:**
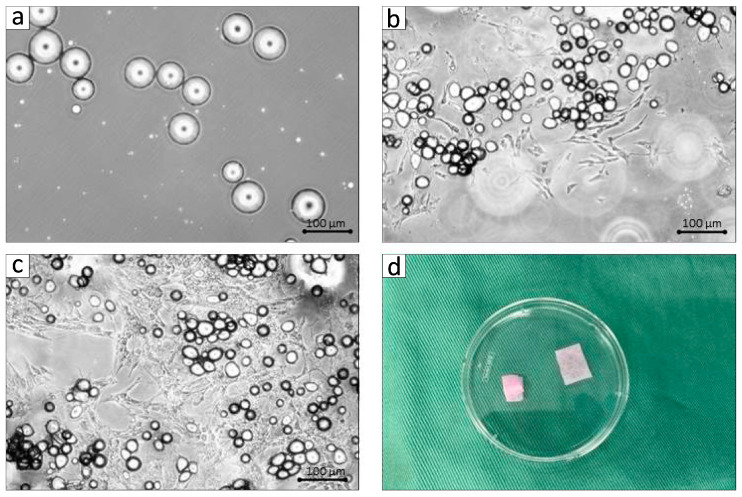
Isolation of DFAT cells and cell seeding. The microscopic examination of primary cultured DFAT cells harvested from the buccal fat pad (BFP) of adult miniature pigs during the ceiling culture process. (**a**) Day 1. (**b**) Day 7. (**c**) Day 10. (**d**) After performing the ceiling culture method, 1.0 × 10^6^ BFP-derived dedifferentiated fat cells are seeded into the collagen matrix for the purpose of transplantation.

**Figure 3 biomolecules-15-00604-f003:**
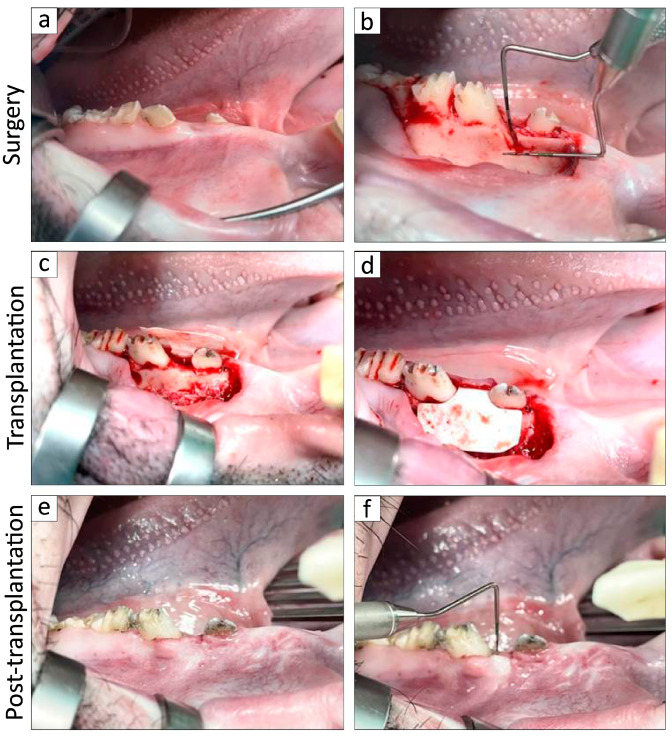
In vivo experiment on the mandible of MMPs. (**a**) Removal of the supragingival calculus of mandibular premolars before surgery. (**b**) The two-walled infrabony periodontal defects (5 mm in height, 4 mm in width, and 3 mm in depth) are formed on the buccal aspect of the bilateral second premolars. (**c**) Four weeks later, the periodontal defect and the buccal bone surrounding the premolars exhibited destruction. (**d**) The collagen matrix was inserted into the defect and subsequently covered with a membrane after periodontal debridement. (**e**,**f**) After twelve weeks, the clinical parameters were reassessed prior to the extraction of the mandible.

**Figure 4 biomolecules-15-00604-f004:**
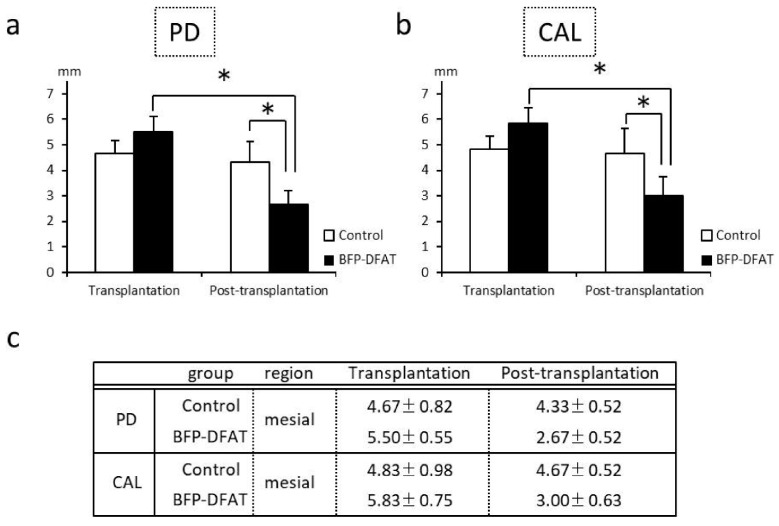
Clinical parameters in the mesial area of the mandibular second premolar at transplantation and post-transplantation. (**a**,**b**) The control group exhibited no significant statistical differences between the transplantation and post-transplantation phases. Conversely, the BFP-DFAT group demonstrated significant differences between these two time points. In addition, a notable difference was identified between the control and BFP-DFAT groups post-transplantation. Each bar in the figure indicates the mean ± SD (*n* = 6, * *p* < 0.05). (**c**) presents the mean values (±SD) for the mesial area of the control and BFP-DFAT groups between transplantation and post-transplantation (mm). The abbreviations used include PD for probing depth and CAL for clinical attachment level, with BFP-DFAT referring to buccal fat pad-derived dedifferentiated fat cells.

**Figure 5 biomolecules-15-00604-f005:**
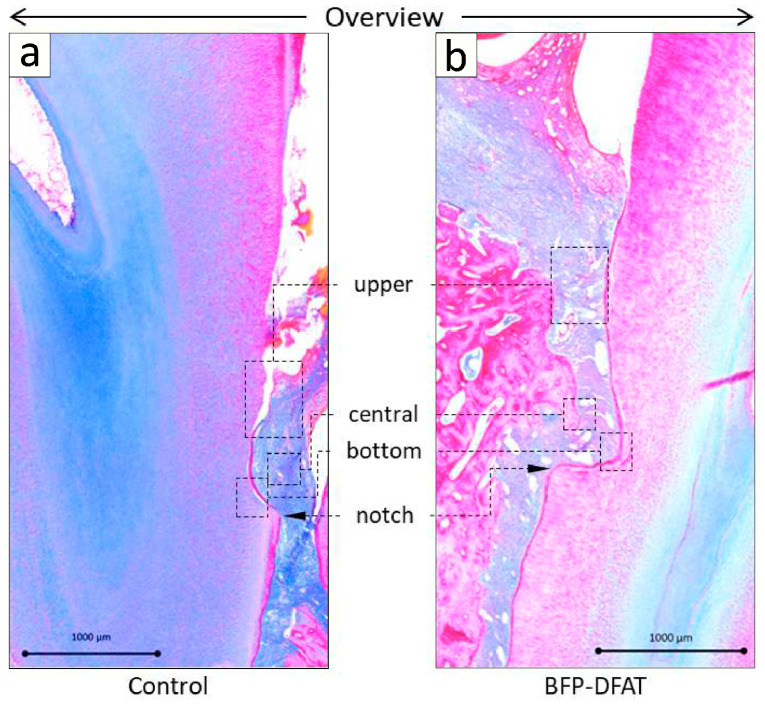
Representative azan-stained histological observation of sagittal plane sections on the mesial root of the second premolar post-transplantation. The two-walled infrabony periodontal defects and notch-shaped marks in the (**a**) control group and (**b**) the BFP-DFAT group are readily discernible at the edge of the images.

**Figure 6 biomolecules-15-00604-f006:**
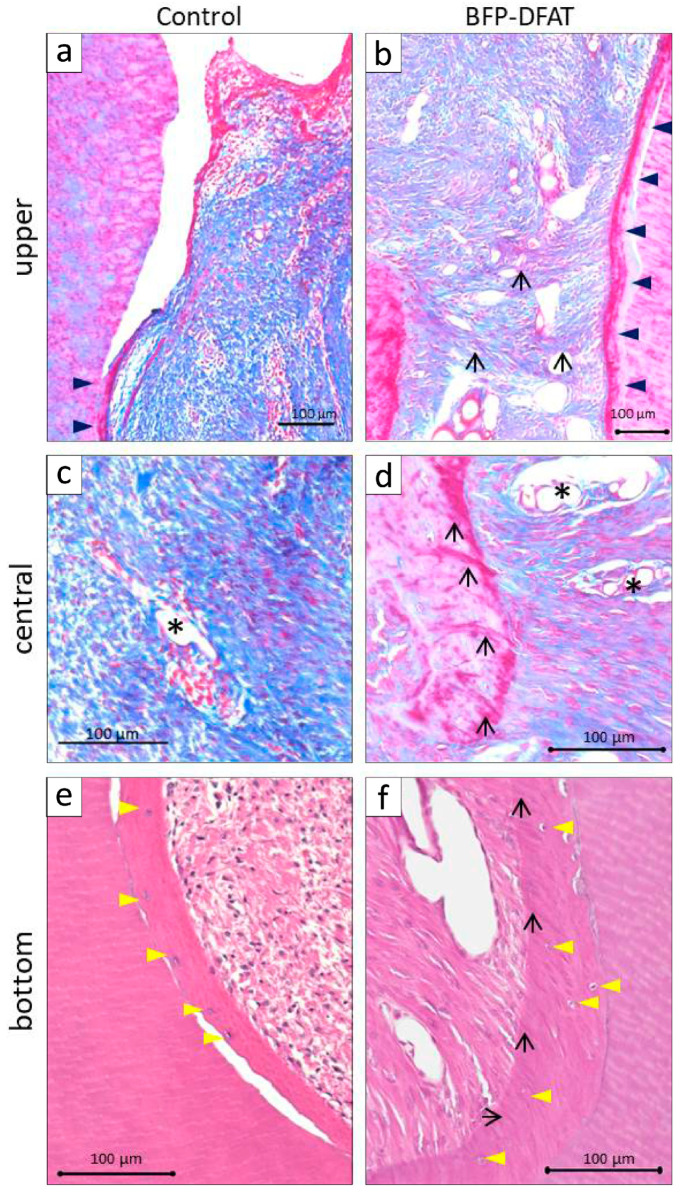
High-magnification images of two-walled infrabony periodontal defect regions from Figure 5. (**a**) Control group: the black arrowheads indicate newly formed hard tissue that has emerged on the exposed dentin surface in the upper section. (**b**) DFAT group: The black arrowheads similarly indicate the newly developed hard tissue on the upper exposed dentin surface. The black arrows indicate newly formed collagen bundles visualized between newly formed hard tissue structures. (**c**) Control group: the asterisk indicates newly formed vascularization in the central section. (**d**) DFAT group: The asterisks indicate newly formed vascularization in the connective tissue. The black arrows indicate collagen bundles inserted perpendicular to the surface of the newly formed hard tissue in the central section. (**e**) Control group: the yellow arrowheads indicate the nucleus of the eosinophilic hard tissue structure on the exposed dentin surface in the bottom section. (**f**) DFAT group: Numerous collagen bundles inserting diagonally (black arrows) into the eosinophilic hard tissue are visualized, which contains hematoxylin-positive cells (yellow arrowheads).

**Figure 7 biomolecules-15-00604-f007:**
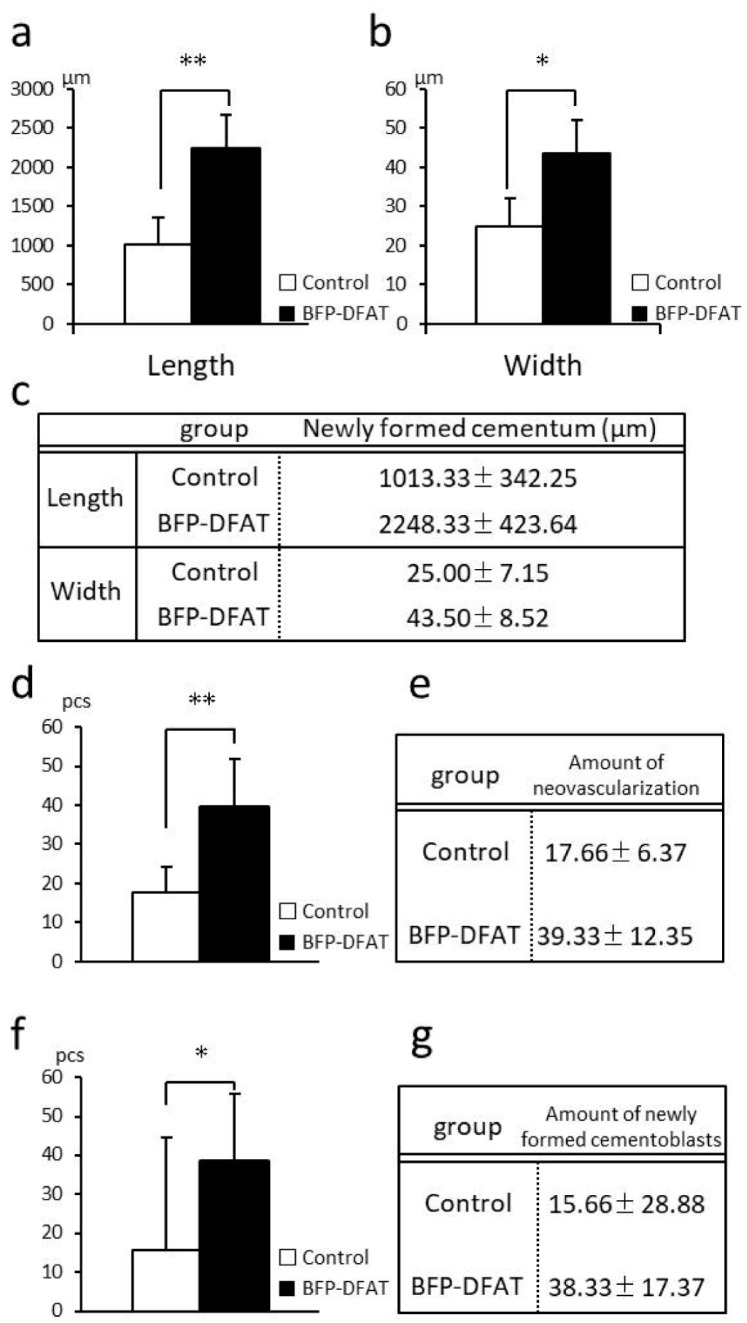
Histometric analysis of newly formed structures within the artificially created periodontal defect. Each bar in the accompanying figures represents the average values ± SDs (*n* = 6, ** *p* < 0.01, * *p* < 0.05). (**a**) Length of cementum measured from the notch to the bottom of gingival sulcus is significantly greater in the BFP-DFAT group compared to the control group. (**b**) Width of cementum from the notch to the bottom of gingival sulcus is also significantly greater in the BFP-DFAT group relative to the control group. (**c**) Results of the average values (±SDs) of the control and BFP-DFAT groups (µm). (**d**) The degree of neovascularization is significantly higher in the BFP-DFAT group when compared to the control group. (**e**) Results of the mean values (±SDs) of the control and BFP-DFAT groups (pieces). (**f**) The quantity of newly generated cementoblasts is markedly increased in the BFP-DFAT group in comparison to the control group. (**g**) Results of the mean values (±SDs) of the control and BFP-DFAT groups (pieces).

**Figure 8 biomolecules-15-00604-f008:**
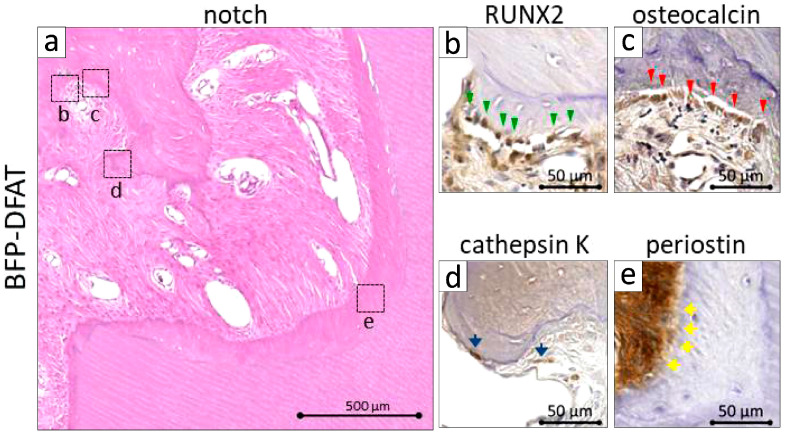
Magnified notch regions of the BFP-DFAT group in Figure 5. (**a**) H&E-stained photomicrographs indicate neovascular structures and countless bundles between the newly formed alveolar bone and cementum. (**b**) At a higher magnification of the dotted frame in (**a**), immunocytochemical analysis demonstrates the presence of RUNX2-positive cells situated along the newly developed alveolar bone. Green arrowheads indicate osteoblasts. (**c**) The immunocytochemical photomicrographs illustrate osteocalcin-positive osteoblasts, which are indicated by the red arrowheads, lining the newly formed alveolar bone. (**d**) Additional immunocytochemical photomicrographs reveal cathepsin K-positive osteoclasts, as indicated by the blue arrows, located in proximity to the newly formed alveolar bone. (**e**) Immunocytochemistry photomicrographs. Yellow arrows indicate periostin-positive periodontal ligaments combined with the newly formed cementum.

**Figure 9 biomolecules-15-00604-f009:**
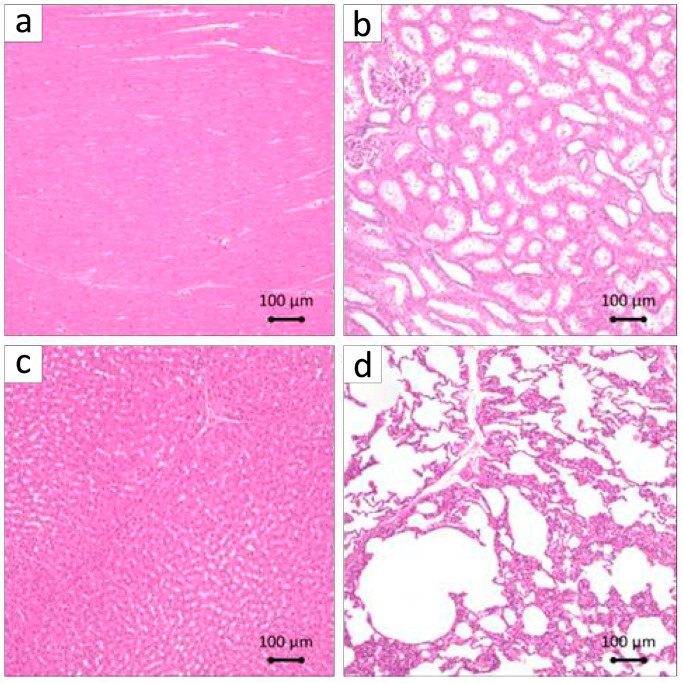
Representative H&E-stained histological observation of internal organs lacking teratomas post-transplantation. (**a**) Heart. (**b**) Kidney. (**c**) Liver. (**d**) Lung. Some inflammation is observed.

## Data Availability

The data presented in this study are available in the article.

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
