# Peer review of "Regeneration of Two-Walled Infrabony Periodontal Defects in Swine After Buccal Fat Pad-Derived Dedifferentiated Fat Cell Autologous Transplantation"

_biomolecules, 2025, doi:10.3390/biom15040604_

Round 1
Reviewer 1 Report
Comments and Suggestions for Authors
This article evaluates the regenerative capacity of periodontal tissue in adult pigs through autologous BFP-DFAT cell transplantation, providing a new cell source for periodontal regeneration. The convenience of BFP sourcing makes it suitable for oral surgical procedures, which is of significant clinical relevance.
However, please consider the following issues:
1. It is recommended that the title reflects the autologous source.
2. It is suggested to include key data in the abstract, such as a 2.2-fold increase in new alveolar bone length and a 2.2-fold enhancement in vascularization.
3. This preclinical study only compares against a standalone collagen matrix; it is advisable to consider adding a positive control, such as collagen matrix combined with other proven effective cells (e.g., MSCs).
4. It is recommended to clarify the randomization method and blinding protocol used in grouping.
5. The current study primarily relies on histology; it is suggested to supplement with Micro-CT imaging for multidimensional validation and to quantify bone volume/density.
6. Vascularization is inferred solely through H&E staining without using specific markers such as CD31.
7. The exploration of mechanisms in this preclinical study may be insufficient; consider adding qPCR detection of osteogenic/dentinogenic markers (e.g., ALP, Runx2) or providing a more detailed analysis of specific mechanisms in the discussion section.
8. The references cited are relatively old and may need to be updated with recent related studies, especially regarding the advancements in BFP-DFAT or DFAT research.
9. There is insufficient comparison of the differences between BFP-DFAT and subcutaneous adipose DFAT, which weakens the persuasive power of the "sourcing advantage"; further discussion on this topic and the feasibility of human application is recommended.
Author Response
Thank you very much for your thorough review.
Comment1→We have added to the title.
Comment2→We have changed the expression.
Comment 3→In this manuscript, we wanted to emphasize the usefulness of BFP-DFAT, including its accessibility, so We did not compare it with other MSCs that have challenges in establishment, but we'll plan to consider in the future.
Comment 4→We have added a note. In reality, we used a smartphone app to assign participants to the BFP-DFAT group and the control group.
Comment 5→We wanted to evaluate hard tissue regeneration using micro-CT, but we were unable to do so due to equipment failure. In future research, we would like to conduct analyses using micro-CT as well.
Comment 6→Thank you for your valuable advice. We would like to utilize it in the evaluation methods of our future research.
Comment 7→We have added it to the discussion.
Comment 8→We have rewritten the discussion, so the references have changed, but we couldn't find much literature on BFP-DFAT.
Comment 9→The points reviewer pointed out are one of the most important issues for our future research. Therefore, we have rewritten the discussion.
Reviewer 2 Report
Comments and Suggestions for Authors
Regeneration of two-walled infrabony periodontal defects in 2 swine after buccal fat pad-derived dedifferentiated fat cells 3 transplantation
This study investigates the potential effects of regenerating periodontal pockets in second premolars using mature adipocyte-derived fat stem cells in swine. The results after 12 weeks reported periodontal attachment. There were previous reports of the same authors performed on the rats.
The introduction clearly states the current knowledge of the issues related to periodontal regeneration using fat stem cells. The use of abdominal and buccal fat cells for regeneration is interesting and can be used in current clinical therapy.
Methods. The study was performed on 6 swine. The technical preparation of the fat cells procedure is clearly described. The periodontal defects were created by burr. The artificially created defects were filled with the cells.
My questions: have these pigs had natural periodontitis with bone defects? Because depending on the answer and the presence of periodontal bacteria regenerative success is highly dependent. The idea is that we have a certain behavior in periodontal-free bacteria tissue and another one in inflamed periodontium due to bacteria…. This must be cleared! Moreover, the bacteria highly modify the tissular reactivity! - in the discussion section
Results- good presentation and comprehensibility both as images and statistics.
Discussion- we understand that the techniques are successful but again the conditions of bacteria-free situ must be discussed and compared with similar cases with bacteria-infected situ. Similar studies of fat stem cells used in periodontics should be presented for a better understanding of the value of results!
Conclusions are fine and in line with the objectives
Author Response
Thank you for your thorough review.
We have incorporated the points you raised into the discussion.

Reviewer 3 Report
Comments and Suggestions for Authors
Please see the attachment.

Author Response
Thank you for your thorough review.
We have incorporated your comments into the summary and introduction sections.
We made some revisions to the expression in the materials and methods.
Regarding reviewer's question about the protocol, based on previous studies, we defined the duration of bacterial infection as four weeks.
Similarly, we determined the duration for regenerative healing to be twelve weeks for evaluation.
We have rewritten the discussion, which includes references to the inflammation observed in the lungs.
We appreciate reviewer's advice.

Round 2
Reviewer 2 Report
Comments and Suggestions for Authors
My concerns have been addressed. Thank you!
Author Response
Thank you very much for your review.We have added some literature regarding periodontal disease in mini pigs to part of the discussion.

Reviewer 3 Report
Comments and Suggestions for Authors
Since not all the requested clarifications were provided by the authors or the answers were comprehensive, I report the necessary clarifications to be answered point by point:
1. Which is the null hypothesis and which the alternative hypothesis both should be written in clear form at the end of the introduction.
2. Following requests the authors added this sentence ‘such as extracting if crown of primary teeth were present or changing the cleaning methods’. At this point it is obviously necessary to specify what the cleaning methods were otherwise the protocol used is not reportable!
3. In the previous review round I had requested ‘Specify how the randomisation of defects in the control or test group was done’. The authors did not specify the randomisation method.
4. The authors replied "Regarding reviewer's question about the protocol, based on previous studies, we defined the duration of bacterial infection as four weeks. Similarly, we determined the duration for regenerative healing to be twelve weeks for evaluation.". A valid reference for both defined points must be added to the manuscript. Otherwise it is a definition not based on a rationale supported by reference.
5. As requested in the previous review round. In the results, the authors should describe and report more data on the inflammation found in the lungs.
I request the authors to provide point-by-point answers in the next review round, as required by the review process, and not just the revised manuscript with revisions.
Author Response
Thank you for reviewing it carefully.
Comments 1: Which is the null hypothesis and which the alternative hypothesis both should be written in clear form at the end of the introduction.
Response 1: We set up the null hypothesis and the alternative hypothesis, and we described them at the end of the introduction.
“Next, we set the null hypothesis that there is no difference in the effects of BFP-DFAT cells transplantation. Conversely, we set the alternative hypothesis that there is an effect from BFP-DFAT cells transplantation.
Specifically, in addition to clinical evaluations such as probing depth (PD) and clinical attachment level (CAL), we examined the effectiveness of BFP-DFAT cells autologous transplantation through histological and immunohistochemical analyses and evaluations. The significance level was established at either 1% or 5%. If the test results show a significant difference, the null hypothesis will be rejected.”
Comments 2: Following requests the authors added this sentence ‘such as extracting if crown of primary teeth were present or changing the cleaning methods’. At this point it is obviously necessary to specify what the cleaning methods were otherwise the protocol used is not reportable!
Response 2: We have added the materials used to the materials and methods section.
“The in vivo experimentation was conducted in accordance with previously established protocols, with minor modifications, such as extracting if crown of primary teeth were present or changing the cleaning methods using sponge brushes and oral care wet wipes (Asahi Group Foods Company, Tokyo, Japan)”
Comments 3: In the previous review round I had requested ‘Specify how the randomisation of defects in the control or test group was done’. The authors did not specify the randomisation method.
Response 3: We described the randomization of samples on a PC in the materials and methods section.
“Subsequently, the flaps were carefully adjusted and fastened using artificial dissolvable sutures (VICRYL; Ethicon Inc., NJ, USA), and the resulting defects were subjected to blind randomization via computer algorithms and subsequently allocated to either the control group or the BFP-DFAT treatment sites.”
Comments 4: The authors replied "Regarding reviewer's question about the protocol, based on previous studies, we defined the duration of bacterial infection as four weeks. Similarly, we determined the duration for regenerative healing to be twelve weeks for evaluation.". A valid reference for both defined points must be added to the manuscript. Otherwise it is a definition not based on a rationale supported by reference.
Response 4: We have added references that established the protocol to the manuscript.
“Four weeks post-surgery, clinical evaluations of PD and CAL were conducted on the mesial root of the second premolar utilizing a periodontal probe (Sun dental Co., Ltd., Osaka, Japan) based on previous study [15,16,29,43,44].”
“Twelve weeks post-transplantation, the evaluations of PD and CAL were conducted once more prior to the bilateral mandible extraction, as depicted (“Post-transplantation”; Figure 3e, f) based on previous study [15,16,29,43,44]”
Comments 5: As requested in the previous review round. In the results, the authors should describe and report more data on the inflammation found in the lungs.
Response 5: We have described the findings observed in the lungs in detail within the results.
“To investigate the formation of teratomas, organs were extracted and stained with H&E. Teratomas were absent in the internal organs, specifically the heart, kidneys, liver, and lungs; however, there was a tendency for capillary congestion in the lung interstitium, with mild lymphocyte aggregation observed in the surrounding area (Figure 9a-d).”
We have answered each item in bullet points. Thank you very much.
Round 3
Reviewer 3 Report
Comments and Suggestions for Authors
Well done!